# Assessment of Parameters to Apply Osmotic Dehydration as Pretreatment for Improving the Efficiency of Convective Drying of Cassava Bagasse

José Serpa-Fajardo [1,2,*], Diego Narváez-Guerrero [2], Carolina Serpa-Padilla [2], Elvis Hernández-Ramos [2] and Gregorio Fernández-Lambert [1,*]

1 Postgraduate Studies and Research Division, National Technological Institute of Mexico/Misantla Higher Technological Institute, Misantla 93821, Mexico
2 School of Engineering, University of Sucre, Sincelejo 700001, Colombia
* Correspondence: jose.serpa@unisucre.edu.co (J.S.-F.); gfernandezl@itsm.edu.mx (G.F.-L.)

**Abstract:** This manuscript corresponds to exploratory research carried out to evaluate and define the operating parameters for applying osmotic dehydration (OD) as pretreatment for efficiency improvement in the convective drying of cassava bagasse, providing a methodology to apply these combined techniques, not only for cassava bagasse but also to serve as a reference for its use in other high-moisture agro-industrial residues. Operating factors significant for moisture loss during OD were determined through sieving analysis. Adequate levels and operating conditions were determined for OD and subsequent drying (osmotic agent concentration, solution-to-sample ratio, agitation speed, immersion time, required draining mesh, draining time, drying temperature, and drying air speed). It was established that an osmotic treatment was able to reduce the moisture of cassava bagasse from 87.7% to 28.94% and subsequent drying time by approximately 38.8%. These results allow the use of these combined techniques for drying cassava bagasse, which in turn, open a research agenda for its application in other agro-industrial residues, contributing to its handling, revaluation, and development of a sustainable agro-industry.

**Keywords:** efficiency; agri-food residue; handling; economy; combined methods

## 1. Introduction

Drying is a unitary operation used to dispose of water contained in a product aiming at the conservation, handling, and/or adaptation of a product for sale or further operation. However, this operation itself leads to considerable energy expenditure, to a greater or lesser extent, depending on solid structure and behavior when it is subject or exposed to certain air conditions as a drying agent. Therefore, in the last decade, the academic and scientific community has reported the usage of combined drying techniques for improving the efficiency of the conventional convective drying process in terms of processing time and energy expenditure. Within these combined techniques, employed for this aim, the application of osmotic dehydration (OD) [1–5]. Dehnad et al. [6] point out that drying is frequently used for food processing and has an impact on their functional properties, as well as on the commercial value and application scope, stating that there is no ideal or optimal technique for drying all products. Before selecting the process to employ, it is necessary to consider factors such as: type of product to dry, desired final functional properties, product susceptibility to heat, and process cost.

OD can be defined as 'a process consisting of partial water removal by immersion of plant tissue in a hypertonic solution' [7]. This operation causes a reduction in the initial food moisture content, which regenerates the partial water loss due to the high osmotic pressure of the hypertonic solution that provides a driving force for the water diffusion from the food tissues to the solution. With it, it is possible to obtain benefits such as: the

inhibition of enzymatic browning, a greater amount of volatile compounds after process extension, the retention of natural color without adding sulfites with no phase changes involved, and therefore, with less energy expenditure [5]. The efficiency of this process is affected by various factors, including: the molecular weight of the osmotic agent; type of osmotic substance used as well as its concentration and temperature; process; relationship, geometry, and size of the raw material; and pressure at which the process is carried out [8]. There are different types of osmotic agents used in the OD process, such as sucrose, lactose, maltodextrin, fructose, glycerol, sorbitol, sodium citrate, sodium chloride, high molecular weight carbohydrates (starch), and even combinations among all of them. Being sucrose and sodium chloride the most used given their advantages above others, such as low energy requirements, easy availability, and low cost [9]. On the other hand, Sodium Chloride is mainly used in OD for the treatment of meat, fish, and vegetables due to its high capacity to reduce water activity [10].

Applying OD as pretreatment before the drying process provides a reduction in drying time and processing costs, largely due to the fact that during the OD process, water is partially removed from the product, causing a significant reduction in its initial moisture content, requiring less subsequent drying time, which ends up into a lower process cost [11].

Currently there is an evident issue with the large amounts of cassava bagasse. It has been estimated that for every 250–300 tons of processed cassava root, around 280 tons of bagasse with 85% moisture are produced [12]. Its high water content makes it highly susceptible to the spread of microorganisms and uncontrolled fermentation, generating a source of constant contamination and bad odors, affecting both, processing plants and surrounding regions.

Cassava bagasse has approximately 16% dry matter made up of up to 82.85% carbohydrates, with reports of starch content greater than 50%, fiber percentage between 15 and 50%, and low protein and ash content [13–16].

Regarding the use of cassava bagasse, research has been carried out in the area of biotechnology, food, biomaterials, composting, and the disposal of pollutants in water sources [17–21].

Among the distinct uses or applications presented by bagasse at the biotechnological level, biofuels' synthesis and production (ethanol, n-butanol, biodiesel, isopropyl alcohol, etc.), bioconversion to obtain products of greater biological value, and production of xanthan gum, lactic acid, and carotenoid pigments stand out. Whereas its application in biomaterials provides alternatives for using this residue by means of developing eco-intelligent and active products [22]. In the food industry, cassava bagasse is mainly used as raw material in animal feed [23].

Despite the different studies reported regarding the use of cassava bagasse, the large volumes generated and its high moisture content without any prior treatment of dehydration or drying, make this residue an unmanageable product to be employed in the different options or possibilities of use. Consequently, along with advances in the use studies, it is very important to similarly advance in studies of efficient alternatives for drying, resulting in less operation time and less energy expense.

OD as a pretreatment operation applied to the drying of food matrices, depending on product structure, hardness degree, and porosity, has made it possible to reduce moisture content in ranges between 15% and 44%, and even 70% in some cases, with reductions in subsequent drying times of between 20% and 42.8% [24–26]. Nonetheless, there is no scientific literature on the application of these combined techniques in the drying of cassava bagasse. For this reason, this study aims to evaluate and define the appropriate operating conditions for the application of the OD process as a pretreatment to improve the time efficiency of convective drying of cassava bagasse. The results of this article provide a reference methodology for the application of these combined drying techniques in cassava bagasse and other high-moisture agro-industrial residues, contributing to its management, revaluation, and the development of a sustainable agro-industry.

## 2. Materials and Methods

### 2.1. Location and Raw Material

This research study was carried out at the pilot plant of the University of Sucre, Puerta Verde branch, located at kilometer 7 of the Sincelejo-Sampués road. The cassava bagasse used was collected at the Almidones de Sucre S.A.S company (Morroa, Colombia), located at kilometer 4.5 of the Sincelejo-Corozal road. The working parameters correspond to the result of a technological surveillance study regarding the application of OD as a unitary operation and OD as a pretreatment of convective drying.

### 2.2. Proximal Characterization of Cassava Bagasse

The determination of moisture, ash, fat, fiber, and protein content was carried out based on the standards 977.11 AOAC [27] oven method, 942.05 AOAC [27] general method, 920.39 AOAC [27] Soxhlet method, 962.09 AOAC [27] gravimetric chemical method, and 955.04 AOAC [27] Kjeldah method, respectively, while the carbohydrate content was obtained by the difference from the equation proposed by [28].

### 2.3. Osmotic Dehydration (OD) of Cassava Bagasse

The operating parameters and levels to be assessed were initially selected according to the scientific literature consistent with a technological surveillance study referring to the application of OD as a pretreatment of the convective drying of different agro-industrial products and later adjusted according to the particular behavior of bagasse when it is subjected to this type of processes.

The OD operation was performed following the methodology described by Estrada et al. [29] and Huamán et al. [30]. Samples of 300 g of fresh cassava bagasse were taken, and osmotic solutions were prepared with concentrations of 25%, 50%, and 75% NaCl; then, the mixture of cassava bagasse-osmotic solution was made considering the relative solution to sample ratio of 4:1, 6:1, and 8:1 respectively. Samples were constantly shaken during the entire immersion time. The stirring speed in revolutions per minute used was defined experimentally based on the need for the mixture to remain homogeneous under environmental conditions of 1 atm and approximately 30 °C; testing stirring speeds between 250 rpm and 500 rpm. The OD operation time evaluated was 1.5, 3, and 4.5 h. The moisture of the samples was determined according to the oven method 977.11 AOAC [27]. A drainage water test was performed to determine the mesh number and adequate drain time for the sample. For this, solution loss tests as a function of time were carried out, working with Tyler sieves with different numbers and mesh openings. Mesh number and draining time were determined from the capacity with which the sieves allowed the passage of the solution and efficient retention of solids or bagasse. For this, Tyler #30, #20, and #10 scale sieves with mesh openings of 0.595 mm, 0.841 mm, and 2 mm, respectively, were used. Next, 350 g of cassava bagasse were taken per sample and spread over each of the sieves; then they were submerged in a container with the osmotic solution for the time defined for the OD. At the end of this time, they were removed and allowed to drain until no water migration was observed through the sieve bottom (approximately 2 min). Finally, sieves were weighed to determine the solution loss in each sample at time intervals of 10, 15, 20, 25, and 30 min.

After the draining operation, samples were carefully treated with absorbent paper, to then measure the moisture by the oven method according to the AOAC [27].

The factors significantly influencing moisture loss during the OD process were determined from a sieving design 23 corresponding to 3 factors with the respective maximum and minimum levels selected. Factor 1: Osmotic agent concentration: 25% and 75% NaCl; Factor 2: Solution to sample ratio: 4:1 and 8:1 g solution/g sample; Factor 3: Immersion time: 1.5 h and 4.5 h. Data processing was carried out through the STATGRAPHICS Centurion XIX software v. 19.1.01, with a 95% confidence.

### 2.4. OD as Pretreatment for Convective Drying of Cassava Bagasse

The OD procedure as a pretreatment to improve the efficiency of convective drying of cassava bagasse was carried out based on the methodology used by Sanez Falcon [31]. Once the OD and draining operations were completed, the samples were taken to the drying chamber previously stabilized in the working conditions. To evaluate the best operating conditions, a 2 × 2 × 3 factorial experiment was developed in a completely randomized design. Factors evaluated were: Factor 1: Osmotic agent concentration (30% and 60% NaCL) and Factor 2: Solution to sample ratio (3:1 and 5:1 g of solution/g of sample). Factor 3 considers that one of the benefits of applying combined drying techniques is the requirement of lower operating temperatures that lead to lower energy expenditure where air speed as a drying agent plays an important role in the efficiency of the drying process, especially when crusting occurs on the surface of the sample due to the use of high temperatures and high concentrations of osmotic agent. In this study, three conditions of the inversely proportional relationship of temperature-air speed were evaluated: 45 °C—1 m/s; 40 °C—1.5 m/s and 35 °C—2 m/s, defined in this way as a first study to explore the behavior of cassava bagasse when applying the OD operation as a pre-treatment to its convective drying, which leads to knowing adequate operating conditions. As a response variable, drying time was evaluated until the samples reached 12% moisture on a wet basis. Under the drying conditions, a sample thickness of 0.005 m was used. The drying operation was performed using a laboratory scale tray dryer TD-S/EV-ELECTTRONICA VENETA, TREVISO, ITALY, Built in AISI 304 stainless steel (Figure 1). It consists of a 450 × 450 × 450 mm drying chamber, which contains four 400 × 300 × 68 mm stainless steel trays for a maximum load capacity of 3500 g. It has a PID (Proportional Integral-Derivative) air speed control system, with a three-phase motor (power of 0.3 Kw, maximum speed of 1400 rpm, and maximum flow of 3.1 m3/h), a stainless steel air filter with a maximum speed of air in the tunnel of 6.6 m/s, a frequency converter to control the speed of the fan, nine 300 W electrical resistors, a thyristor to control the power of the resistors, an electronic scale with digital display (with divisions of 0.1 g), two combined temperature/humidity transmitters (scales from −40 to 60 °C and 0 to 100% relative humidity with divisions of 0.1 °C and 0.1% respectively); one digital anemometer (scales from 0 to 10 m/s) and an IP 55 electrical panel (with synoptic table, three electronic temperature display screens, two relative humidity, fan, resistance switches, and potentiometers). All the tests were carried out in triplicate and compared with a standard treatment that consisted of drying a sample of bagasse without prior osmotic treatment. The moisture of the samples was determined every 40 min, after 5 h of operation, using the AOAC 977.11 oven method [27].

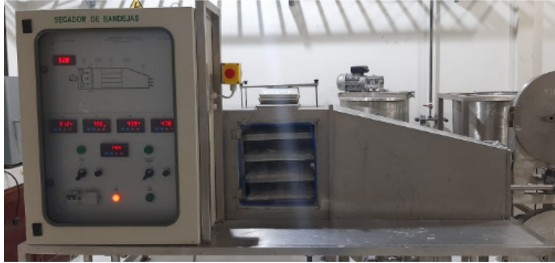

**Figure 1.** Drying Equipment.

## 3. Results and Discussion

### 3.1. Proximal Characterization

The results of the proximal characterization are presented in Table 1, the cassava bagasse presented a carbohydrate content greater than 70%, with a low-fat content similar to that reported by Romero de Armas et al. [32]; likewise, the amount of fiber does not contrast significantly with the results presented by Paternina et al. [33]. However, it presents a difference greater than 20% with the fiber content present in the bagasse characterized by Polachini et al. [14]. The differences present in the composition of cassava bagasse from

different sources are due to factors such as the type of process, whether it is artisanal or industrial, the efficiency of the starch extraction process and peeling of the raw material, Bussolo de Souza et al. to the [15].

**Table 1.** Proximal characterization of cassava bagasse.

| Components | Contents |
|---|---|
| Moisture (moist basis) | $87.877 \pm 0.314\%$ |
| Dry matter (moist basis) | $12.123 \pm 0.315\%$ |
| Ashes (dry basis) | $1.6310 \pm 0.017\%$ |
| Protein (dry basis) | $1.8840 \pm 0.250\%$ |
| Fat (dry basis) | $0.5120 \pm 0.075\%$ |
| Fiber (dry basis) | $24.692 \pm 0.277\%$ |
| Carbohydrates (dry basis) | $71.281 \pm 0.468\%$ |

### 3.2. Stirring Needs during OD

Bagasse behaviour immersed in the different samples without agitation showed that a standardized constant agitation was required to guarantee a homogeneous mixing of the working mixture, and in turn, guarantee the reliability of a good sampling process resulting in a correct measurement of moisture as a response variable.

Figure 2A reveals a phase split when the sample is not shaken. The first phase corresponds to the less dense fraction of the bagasse; followed by a fraction of the diluted osmotic solution. Then the densest fraction of the bagasse was located and finally, the osmotic agent used was found, evidencing the need to apply adequate agitation applied constantly throughout the cassava bagasse OD process, considering that low levels of agitation do not allow a homogeneous mixture of its constituents. Most of the studies involving the OD technique are consistent with a single level of agitation [2,3,34–36]. This is due because very high speed differences are needed to be able to appreciate the effect of this variable in the OD process.

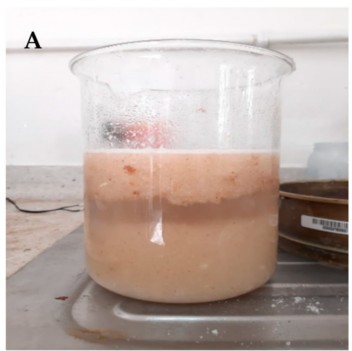 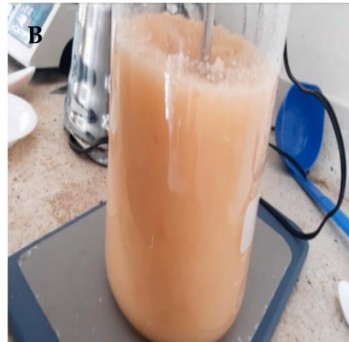

**Figure 2.** OD at 75% NaCl and solution to sample ratio of 4:1; (**A**) without agitation; (**B**) stirred at 320 rpm.

Figure 2B shows the behavior of an OD test at 75% NaCl, with a 4:1 ratio and stirring at 320 rpm, a stirring speed that made it possible to obtain a homogeneous mixture throughout the entire used container.

After carrying out several stirring tests between ranges of 250–500 rpm, it was determined to use a stirring speed of 320 rpm as the speed that guarantees homogeneity or uniform mixing of the solution.

### 3.3. Adjustment of Osmotic Agent Concentration Levels and Solution to Sample Ratio

After carrying out the OD test for the treatment with a concentration of 75% NaCl and a solution-to-sample ratio of 4:1 and 8:1, an exaggerated amount of salt as an osmotic agent and a small amount of cassava bagasse were evident in the samples (Figure 3A,B). Due to

the morphological structure of cassava bagasse, because of its minute particle size, it allows a considerable gain in solids, thanks to its greater surface area [37].

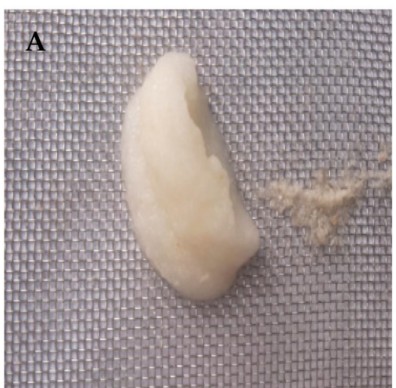 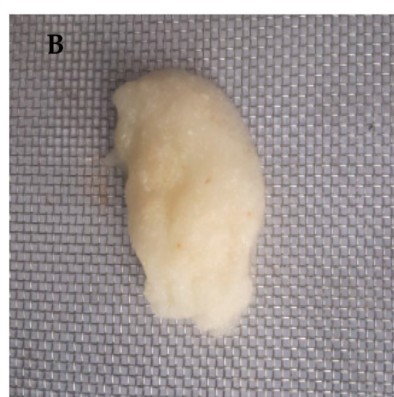

**Figure 3.** OD samples; (**A**) 75% NaCl and 4:1; (**B**) at 75% NaCl and 8:1.

*3.4. Determination of Mesh Number and Drain Time*

Additionally, high amounts of salt would generate a disadvantage in terms of dry bagasse production costs, in contrast to the aim of achieving the lowest possible production costs with the lowest energy expenditure. Therefore, these operating levels were discarded and adjusted, reducing both the concentration percentage of the osmotic agent and solution to sample ratio.

New operating conditions were established: NaCl concentrations between 30 and 60% and solution-to-sample ratios between 3:1 and 5:1, which would allow the maximum use of levels of fiber, protein, and other components shown by the cassava bagasse [15,32,33].

Results of the weight loss test indicated that the #30 sieve did not allow the solution passage, so it was discarded. Figure 4 illustrates the amount of water retained as a time function in the draining operation. The #10 sieve held 62 g of water, while the #20 sieve held 80 g. Therefore, the sieve selected for the draining operation was #10. Nevertheless, it was observed that sieve #10 lets some bagasse particles escape, which is why sieve #12, corresponding to the number of mesh in its order, was selected as the draining mesh. Furthermore, it was observed that, after 20 min in the draining stage, the amount of retained water did not vary significantly, so it was decided to define this time as the optimum for the operation.

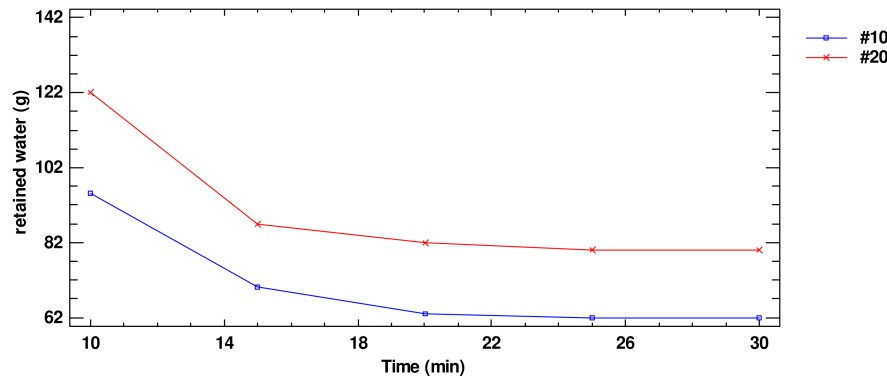

**Figure 4.** Retained water as a function of time.

*3.5. Determination of Significant Factors in Moisture Loss during the OD Operation*

Implementing a draining operation allowed less manipulation of the evaluated samples, especially when executing OD as pretreatment of convective drying. This is because trays were designed so that their bottom had the opening size of the selected mesh. Navarrete [38] demonstrates this, by applying OD on pieces of papaya with a draining time of

30 min prior to convective drying at 60 °C and obtaining as a result, that an excessive loss of nutrients and less handling of the samples was avoided.

To estimate the process factors that significantly influence moisture loss during the OD process, a sieving design with a 23 factorial arrangement was applied for the factors: concentration of the osmotic agent at two levels (30% and 60%), the solution-to-sample ratio at two levels (3:1 and 5:1), and immersion time at two levels (1.5 h and 4.5 h), using the moisture percentage as a response variable. Table 2 lists the results obtained for moisture in the respective sieving test.

**Table 2.** Moisture results according to the sieving test.

| Concentration (%NaCl) | Ratio s:m (g sol/g m) | Time (h) | Moisture (%) |
|---|---|---|---|
| 30 | 3 | 1.5 | 71.794 |
| 60 | 3 | 1.5 | 37.616 |
| 30 | 5 | 1.5 | 69.832 |
| 60 | 5 | 1.5 | 28.449 |
| 30 | 3 | 4.5 | 70.435 |
| 60 | 3 | 4.5 | 37.911 |
| 30 | 5 | 4.5 | 70.019 |
| 60 | 5 | 4.5 | 29.429 |

Fresh cassava bagasse has moistureof 87.7%, when working with a hypertonic solution at 60% NaCl and a solution-to-sample ratio of 5:1, it was possible to bring this moisture mean content up to 28.94%.

Only the osmotic agent concentration and solution-to-sample ratio factors were significant ($p < 0.05$) for bagasse moisture loss during the OD process (Table 3).

**Table 3.** Analysis of Variance for the moisture sieving test obtained in the OD process.

| Source | Sum of Squares | DF | Mean Square | F-Ratio | *p*-Value |
|---|---|---|---|---|---|
| A: Concentration | 2763.01 | 1 | 2763.01 | 29851.85 | 0.0037 * |
| B: ratio s:m | 50.1326 | 1 | 50.1326 | 541.64 | 0.0273 * |
| C: Time | 0.00133903 | 1 | 0.00133903 | 0.01 | 0.9238 |

* significant for a significance level of 0.05.

The standardized Pareto diagram (Figure 5) illustrates the results exposed by the ANOVA, in terms of significance ($p < 0.05$) of the factors concentration of NaCl, solution-to-sample ratio, and interaction. Osmotic agent concentration is the factor that most affects moisture loss of the bagasse when it is subject to a DO process. While the time factor has no significant effect on the studied response variable.

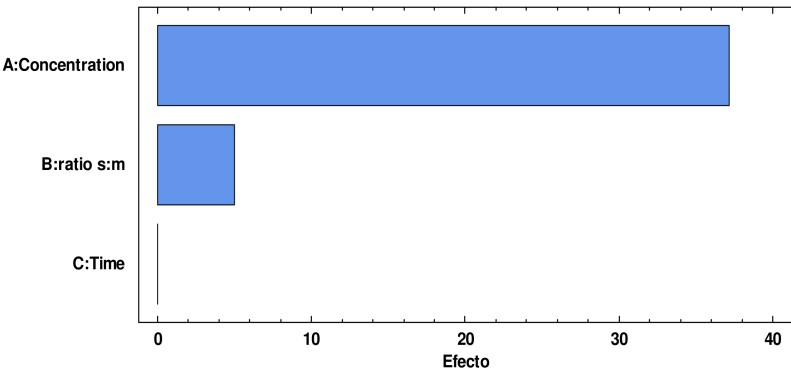

**Figure 5.** Standardized Pareto Chart.

Figure 6, on its part, shows that the concentration of osmotic agent can lower the moisture percentage from approximately 90% (which is the initial moisture value of fresh

cassava bagasse) to values between approximately 71% and 33%. Banbicha et al. [39] managed to reduce the moisture content of pumpkins from about 90% to about 31%, after 3 h of OD in a hypertonic solution with 60% sucrose and 9% NaCl. For their part, Sanez Falcón [31] and García-Paternina et al. [40] obtained moisture values of 12.28 and 14.27% when using concentrations of 60 and 65% sucrose on blueberries and mamey, respectively. Figure 6 also illustrates the incidence of the factor solution to sample ratio on the variation of the bagasse moisture percentage, that although it has a significant effect, it is lower or less marked than the effect given by the osmotic agent concentration and the null effect of the immersion time on the response variable. García-Paternina et al. [40] and Estrada et al. [29], observed reductions in the percentage of moisture in mango and guava samples of up to 10.5 and 5.88%, respectively, using ratios of the solution in a sample of 1:1 and 2:1 sucrose to glucose and sucrose with immersion times between 2 h and 48 h.

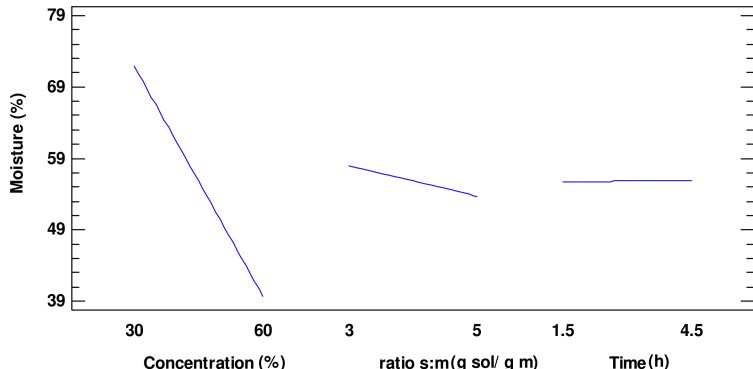

**Figure 6.** Plot of main effects.

### 3.6. Determination of the Optimal Immersion Time

When cassava bagasse is subject to immersion in an OD process under agitation conditions of 320 rpm, bagasse particles are released into the osmotic solution, generating a greater contact surface area that causes greater efficiency of the dehydration process, rapidly reaching equilibrium.

Figure 7 shows that after 5 min of treatment, the bagasse sample loses almost all the possible moisture in all the tests, quickly tending to equilibrium; however, from a time of 10 min, stability can be observed in the obtained moisture values.

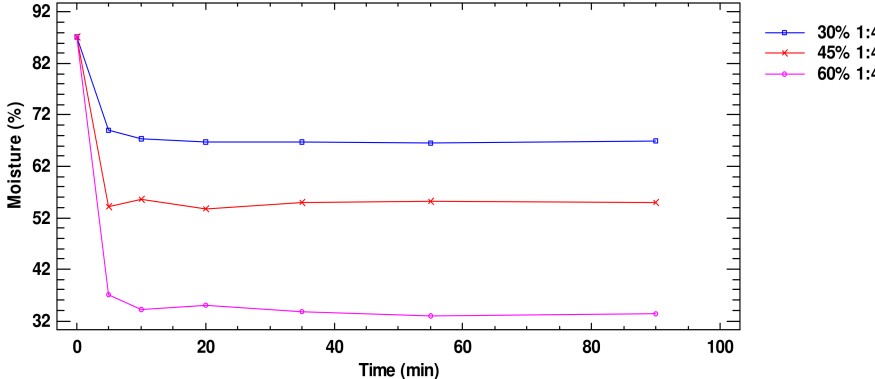

**Figure 7.** Moisture during OD.

Based on this, an immersion time of 10 min was defined and used to carry out all subsequent tests, corresponding to the application of OD as pretreatment of convective drying. In the state of the art immersion times of 1, 1.5, 2, 4, 5, 12, 16, 24, and up to 48 h were found [1,31,39,41–45]. To reach equilibrium, long periods of time are generally needed. Nonetheless, it has been found that mass transfer is not significant after 4 to 5 h [31]. Process

time depends on the changes to be achieved in the raw material: impregnation, osmotic pretreatment, OD, production of candied fruit [31], and its nature, form, and composition.

### 3.7. Behavior of Cassava Bagasse during Application of Combined Methods

Based on the adjustments made to each operating parameter, the use of combined methods using OD as pretreatment was evaluated to improve the efficiency of convective drying of cassava bagasse. Table 4 shows results obtained during the carried-out tests. It was possible to establish that the application of these combined methods is not feasible to improve the drying efficiency of cassava bagasse, at the used concentration levels of NaCl of 30% and 60%, and solution to sample ratios of 3:1 and 5:1 g solution/g sample, under drying conditions with a temperature of 45 °C with an air speed of 1 m/s, and 40 °C with 1.5 m/s, respectively; treatments in which crust formation occurred on the surface of the sample that limited the processes of diffusion, mass transfer and heat transfer, affecting the efficiency of the cassava bagasse drying operation. While the control treatment achieves 12% moisture on a wet basis at 720 min. At that same time, the treatments developed under these operating conditions showed much higher moisture in ranges between 25 and 49% moisture. Similar results were obtained by Taşova et al. [46], who, by subjecting pear slices to OD with 75% sucrose, obtained a drying time even longer than the control.

**Table 4.** Tests with adjusted OD parameters as pretreatment of convective drying.

| Trial | Concentration (%NaCl) | Solution to Sample Ratio (g sln/g m) | Drying Conditions | Drying Time Up to 12% bh (min) | Final Moisture [2] (%) | Processing Time [3] (min) | Observations |
|---|---|---|---|---|---|---|---|
| 1 | 60 | 3:1 | 45 °C 1 m/s | Nf [1] | 34.55 ± 1.98 | 880 | Crust formation that prevents drop of Moisture. At 720 min required by the standard treatment to reach 12% Moisture, samples still show high Moisture values. |
| 2 | 60 | 5:1 | 45 °C 1 m/s | Nf | 32.28 ± 3.21 | 920 | |
| 3 | 30 | 3:1 | 45 °C 1 m/s | Nf | 27.73 ± 2.52 | 840 | |
| 4 | 30 | 5:1 | 45 °C 1 m/s | Nf | 28.52 ± 1.10 | 900 | |
| 5 | 60 | 3:1 | 40 °C 1.5 m/s | Nf | 30.29 ± 2.50 | 880 | |
| 6 | 60 | 5:1 | 40 °C 1.5 m/s | Nf | 26.44 ± 1.37 | 820 | |
| 7 | 30 | 3:1 | 40 °C 1.5 m/s | Nf | 25.29 ± 2.91 | 900 | |
| 8 | 30 | 5:1 | 40 °C 1.5 m/s | Nf | 25.83 ± 1.70 | 880 | |
| 9 | 60 | 3:1 | 35 °C 2 m/s | Nf | 48.84 ± 1.40 | 820 | Insufficient drying conditions |
| 10 | 60 | 5:1 | 35 °C 2 m/s | Nf | 47.96 ± 2.39 | 900 | |
| 11 | 30 | 3:1 | 35 °C 2 m/s | Nf | 42.44 ± 3.15 | 880 | |
| 12 | 30 | 5:1 | 35 °C 2 m/s | Nf | 43.86 ± 0.96 | 920 | |
| | | | | Additional tests, lowering the concentration of osmotic agent: | | | |
| 13 | 25 | 3:1 | 40 °C 1.5 m/s | Nf | 27.47 ± 2.10 | 880 | No drop in Moisture |
| 14 | 20 | 3:1 | 40 °C 1.5 m/s | 440 ± 20.17 | 12.00 | 440 | Excellent Result |
| Pattern | | - | 40 °C 1.5 m/s | 720 ± 15.28 | 12.00 | 720 | None |

[1] It was not possible to report the time required to reach 12% moisture, because in these treatments there was not a sufficient drop in moisture levels. [2,3] They refer, respectively, to the final moisture reached and the required drying time. The process was stopped after observing constant moisture for more than two hours, exceeding the time required by the Pattern to reach a value of 12% moisture.

The crust generated on the surface of the samples, attributed to the use of a high concentration of the osmotic agent and high air temperatures as a drying agent, forms a barrier that prevents the moisture retained inside them from being eliminated by the air under the given conditions of temperature and speed, which impairs the efficiency of the drying operation. This behavior is consistent with the results reported by Della Rocca and Mascheroni [9], who emphasize that the formation of the layer on the evaluated materials prevents moisture migration and weight loss over time.

Figure 8A,B show the formation of this crust on the upper and lower surfaces of the used sieves. Said behavior is given as a consequence of the surface hardening undergone by the crust due to the crystallization of the sugars or compounds contained in the osmotic solution. If the amount of osmotic agent used is too high, as well as temperature and air speed, a superficial crust is formed on the assessed material, which will prevent the entry or exit of both solids, and moisture and will lead to an increase in convective drying time [47].

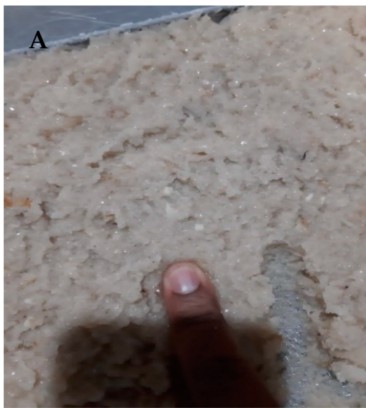 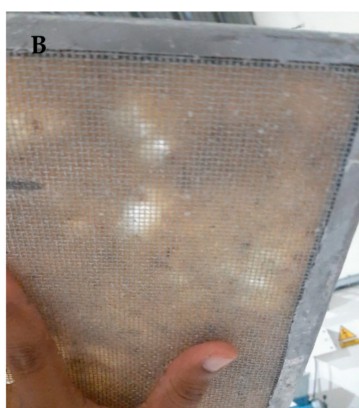

**Figure 8.** (**A**) Crust formation on the upper surface of the assessed samples; (**B**) Crust formation on the lower surface of the assessed samples.

Perez-Won et al. [25] through their research, evidence the development of a layer of concentrated solids under the surface of lemon peels when using a concentration of 70% sucrose (p/p), sample to solution ratio of 20:1, immersion time of 2 h and agitation of 100 rpm while drying conditions were 50 °C and 1.5 m/s. This led to an alteration of the osmotic pressure gradient across the product-medium interface and lowered the driving force for the water flow, which translated into an increase in the osmo-convective drying time (from 5.73 to 6.95 a.m.). In this manner, some authors through their studies show the relationship between the use of high concentrations of osmotic agents, plus high drying temperatures and the increase in drying time. Bomba fruit and jackfruit were treated under working parameters of 75% sucrose and 20% NaCl, respectively, and drying conditions of 60 °C and 2 m/s, and 70 °C and 1.5 m/s, respectively. Obtaining a result increases in the drying time [48–50], and a longer drying time than the control as it has been reported by subjecting pear slices to OD operation processes with a high concentration of osmotic agent (75% sucrose) [50].

For drying temperature conditions of 35 °C and an air velocity of 2 m/s, in the treatments corresponding to 30 and 60% osmotic agent and solution to sample ratio of 5:1 and 3:1, respectively, although there was no crust formation, the conditions of the air used as a drying agent were not enough to overcome the attractive force that existed between the water and the osmotic agent present in the samples, preventing the drop in moisture in said treatments. This is because, according to [48], sodium chloride is a hygroscopic mineral, i.e., it is capable of absorbing moisture from the air and from the material that surrounds it [49], which brings about a change in the colligative properties of the solutions containing it. Colligative properties, on the other hand, are those that only depend on the concentration of the solute and not on the nature of its molecules. These are not related to the size or any other property of the solutes. They are a function only of the number of particles and are the result of the same phenomenon: the effect of solute particles on the solvent vapor pressure. The four colligative properties are: lowering of the vapor pressure of the solvent, ebullioscopic increase, cryoscopic lowering, and osmotic pressure [50]. In the carried out tests, the osmotic agent absorbed the moisture present in the process for itself and did not allow the air used to dry the product.

Given the results obtained from the treatments studied, in which it was not possible to achieve an efficient decrease in the moisture of the cassava bagasse, two additional tests were carried out, decreasing the concentration of the osmotic agent to 25 and 20% NaCL, respectively.

Figure 9 illustrates the curves of moisture vs. time results corresponding to each experimental test carried out. It can be observed that with the exception of treatment 14 in which an excellent result was obtained, achieving the final moisture of 12% in a time of 440 min, in all the other treatments the desired final moisture of 12% was not reached, due to due to the reasons stated above, which in some cases is due to the crust formation that

affects the heat and material transfer processes required for efficient drying, and in other cases due to insufficient drying conditions.

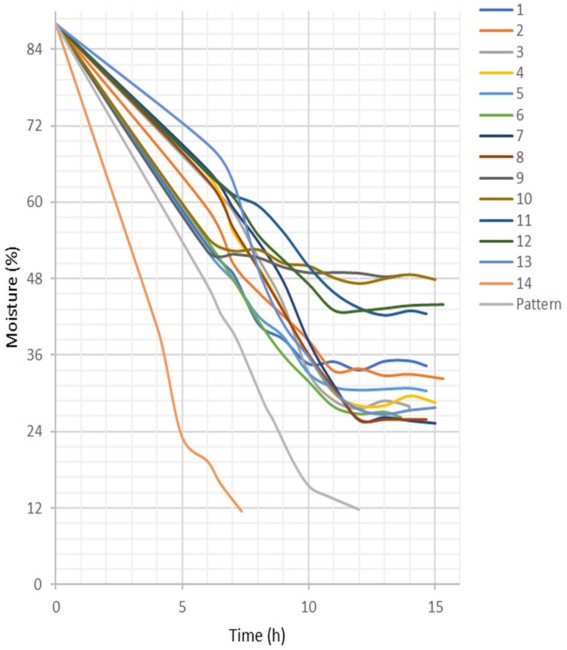

**Figure 9.** Curves moisture vs. time.

It was possible to determine that the application of combined OD techniques as a pretreatment to improve the efficiency of the convective drying process of cassava bagasse is only feasible when working with concentrations of NaCl as an osmotic agent of 20%, with drying conditions of 40 °C temperature and 1.5 m/s air speed.

Using a concentration of 20% NaCl and drying conditions of 40 °C and 1.5 m/s, a reduction in drying time can be achieved, going from 720 min required by the standard treatment to 440 min from said operating conditions. This corresponds to a reduction of approximately 38.8% in the convective drying time of cassava bagasse required to reach a final moisture content of 12%.

## 4. Conclusions

In the cassava bagasse drying operation, when a previous OD operation is used with NaCl as an osmotic agent, low concentrations limited to 20% NaCl should be applied, to avoid the formation of a crust on the surface of the material that affects the efficiency of the processes of diffusion, mass transfer, heat transfer, and drying of this residue.

The parameters and operating conditions were determined for the application of osmotic dehydration as a pretreatment to improve the efficiency of convective drying of cassava bagasse using salt as an osmotic agent.

This study made it possible to establish a methodology for the application of the DO operation as a pretreatment to improve the efficiency of the cassava bagasse drying process, while providing reference parameters for its industrial scaling and application in other high-moisture agro-industrial residues. Contributing to the mitigation of the environmental impact caused by poor waste management and contributing to its management and revaluation for the development of a sustainable agro-industry.

**Author Contributions:** Writing—original draft preparation, J.S.-F.; writing—review and editing, G.F.-L.; conceptualization, E.H.-R.; methodology and formal analysis, J.S.-F., D.N.-G., and C.S.-P. All authors have read and agreed to the published version of the manuscript.

**Funding:** This research was funded by the Ministry of Science, Technology, and Innovation of Colombia, MINCIENCIAS, by supporting two young researchers, under Resolution 0359 of 2021, and the APC was funded by Universidad de Sucre through the publication budget category for projects approved in accordance with the resolution of the Academic Council No. 38 of 2021.

**Institutional Review Board Statement:** Not applicable.

**Informed Consent Statement:** Not applicable.

**Data Availability Statement:** Not applicable.

**Conflicts of Interest:** The authors declare no conflict of interest.

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
