# Peer review of "Assessment of Parameters to Apply Osmotic Dehydration as Pretreatment for Improving the Efficiency of Convective Drying of Cassava Bagasse"

_applsci, doi:10.3390/app122312101_

Round 1

Reviewer 1 Report

The present work presents a study of osmotic dehydration and convective drying of cassava bagasse. The work needs a good improvement before being published in the journal, since the experimental methodology is not properly presented and the analysis of drying data is not well supported. More detailed comments can be found below.

1. Please inform the chemical and physical properties of cassava bagasse.

2. It is common when studying drying is to fix one variable and vary the others. In the work, the authors varied the air velocity at the same time that they also varied the temperature. Knowing that in some cases the effect of velocity is preponderant to that of temperature and that the variation of any of these variables influences the drying kinetics to a certain degree, please provide a well-founded explanation for this.

3. Regarding the experimental design, the authors inform that the drying time was evaluated until the moment in which moisture was equal to 12%. However, when consulting the data in Table 3, the time is equal to 720 minutes for all the conditions. Why does this occur? It is expected that the variation of operating conditions will influence this final time. On the other hand, the moisture is varying. If the authors fixed the drying time, this is not clear in the text and should be corrected.

4. When referring to the solid, the correct technical term is “moisture”, not “humidity”.

5. The experimental methodology is not well described, especially for drying. How was the moisture of the material determined? In addition, the experimental equipment needs to be better detailed. Where were the temperature and air velocity measurements taken? What instruments were used? It is to be expected in works involving drying that, at the very least, these questions need to be answered.

6. Drying kinetic curves need to be demonstrated (moisture versus time) and analyzed. 

7. Please check the moisture units in the graphs in Figures 6 and 7. Check this throughout the text and also in the tables.

Author Response

Consulte el archivo adjunto.

Reviewer 2 Report

ASSESSMENT OF PARAMETERS TO APPLY OSMOTIC DEHYDRATION AS PRETREATMENT TO IMPROVE EFFICIENCY OF CONVECTIVE DRYING OF CASSAVA BAGASSE

 Qusetion one: The second paragraph, the third paragraph and the fourth paragraph of the introduction have no linking words.

 Question two: The background in Figure 3 is best done with white paper.

 Question three: The combination of OD and emerging technologies described in this introduction is not explained.

 Question four: Paragraph 5 and paragraph 11 in the introduction are similar.

 Question five: I think it is necessary to specify the sieves for 15 well sieves.

 Question six: In 3.4, "high amounts of salt would generate a disadvantage in terms of dry bagasse production costs" seems to have been known in advance. Is the concentration of the experimental scheme designed at the beginning too high?

 Question seven:After the draining operation, samples were carefully treated with absorbent paper, to then, measure the humidity by the oven method according to the AOAC.” Does absorbent paper affect the results of the experiment?

 Question eight: NaCl is hygroscopic. How to deal with the inconsistency between the actual concentration of NaCl and the experiment?

Round 2

Reviewer 1 Report

Point 2:  It is common when studying drying is to fix one variable and vary the others. In the work, the authors varied the air velocity at the same time that they also varied the temperature. Knowing that in some cases the effect of velocity is preponderant to that of temperature and that the variation of any of these variables influences the drying kinetics to a certain degree, please provide a well-founded explanation for this.

 Response 2: This manuscript is a first study to explore the behavior of cassava bagasse against the application of the osmotic dehydration operation as a pretreatment to its convective drying, in order to clearly appreciate the effects of the osmotic agent concentration and solution-sample ratio. , used on the response variable (drying time), three conditions of temperature-air speed relationship were established.

PLEASE, INCLUDE A MORE IN-DEPTH EXPLANATION IN THE TEXT FOR THE COMMENTS RAISED IN POINT 2.

Reviewer 2 Report

The manuscript can be accepted.

Author Response

Comments: The manuscript can be accepted

Response: Thank you